# Nitroaromatic Hypoxia-Activated Prodrugs for Cancer Therapy

**DOI:** 10.3390/ph15020187

**Published:** 2022-02-02

**Authors:** William A. Denny

**Affiliations:** Auckland Cancer Society Research Centre, School of Medical Sciences, University of Auckland, Private Bag 92019, Auckland 1142, New Zealand; b.denny@auckland.ac.nz

**Keywords:** hypoxia, nitroaromatics, triggers, reductive fragmentation, one-electron reduction potentials, cancer therapy

## Abstract

The presence of “hypoxic” tissue (with O_2_ levels of <0.1 mmHg) in solid tumours, resulting in quiescent tumour cells distant from blood vessels, but capable of being reactivated by reoxygenation following conventional therapy (radiation or drugs), have long been known as a limitation to successful cancer chemotherapy. This has resulted in a sustained effort to develop nitroaromatic “hypoxia-activated prodrugs” designed to undergo enzyme-based nitro group reduction selectively in these hypoxic regions, to generate active drugs. Such nitro-based prodrugs can be classified into two major groups; those activated either by electron redistribution or by fragmentation following nitro group reduction, relying on the extraordinary difference in electron demand between an aromatic nitro group and its reduction products. The vast majority of hypoxia-activated fall into the latter category and are discussed here classed by the nature of their nitroaromatic trigger units.

## 1. Introduction: The Problem of Tumour Hypoxia

Hypoxia (defined as a free oxygen concentration of <0.1 mMHg) is a common phenomenon in clinical solid tumours [1] and in rodent laboratory models of transplanted tumours [2]. It is caused primarily by the relatively disordered blood capillary networks that develop in rapidly growing tumours, meaning that oxygen can be almost completely consumed before reaching the most distant regions of the tumour. While this is a dynamic and fluctuating situation, significant sections of the tumour at any one time can be severely hypoxic. Such cells can also be more drug-resistant by being out of cycle and by expressing resistance genes under hypoxic stress [3]. Drugs also have to diffuse further from the blood vessels to reach the hypoxic tumour cells. Collectively, this presents a major problem for effective treatment. However, at the same time, hypoxic cells possess a unique environmental difference between themselves and all normal well-oxygenated tissues, and this represents a therapeutic opportunity for a particular class of drugs, the hypoxia-activated “cytotoxins” or “prodrugs” [4,5]. These are comprised of a “trigger” unit attached to a latent cytotoxin which is deactivated when in the prodrug form (Figure 1).

## 2. Prodrugs Activated by Electron Redistribution

The majority of this class of hypoxia-activated prodrug employs a nitroaromatic trigger unit, suitable because of the unique electronic properties of nitroaromatic compounds, which can accept up to six electrons from various reductase enzymes (e.g., NADPH: cytochrome P450 reductase, ferredoxin: NADP+ reductase), and undergo successive formation of identifiable species: nitro radical anion (A; 1 electron), nitroso (B: 2-electron), hydroxylamine (C; 4-electron) and amine (D; 6-electron) species (Figure 1).

The first one-electron step generates the transient nitro radical anion A, which can be efficiently scavenged (re-oxygenated) in a “futile cycle” by sub-micromolar levels of oxygen [6,7], preventing significant further metabolism in well-oxygenated normal tissue. In the hypoxic regions of solid tumours, this re-oxygenation is absent, allowing successive further enzymic reduction of the nitro radical anion to nitroso, hydroxylamine and amine species (B-D) (Figure 1). This cascade greatly increases the electron density at the nitrogen bearing the substituent R, as shown by the Hammett substituent constants (a measure of the electron-donating or -withdrawing ability of a substituent on an aromatic ring); σ_p_ NO_2_ = +0.78 (electron-withdrawing); σ_p_ NH_2_ = −0.66 (electron-donating). This large change in electron density can be exploited for activation of the group R; for example, greatly enhancing the cellular toxicity of nitrogen mustards as DNA crosslinking agents. An early compound of this general class was the dinitroaziridine CB1954 (**1**) (Figure 2), which was shown [8] to have good bioactivity in rat tumours in vivo, due to activation primarily by the reductase NQO1 (DT-diaphorase). The mammalian version of the enzyme is much less active, limiting the further development of **1**, but it remains a useful comparator molecule. A study [9] of substituted *N*-phenyl nitrogen mustards showed a difference of up to 7200-fold in toxicity (measured as CT_10_ = concentration x time to reduce the surviving fraction to 10% of controls) between compounds **2** and **3** in cultured UV4 CHO cells. While this ratio is augmented by the relatively high sensitivity of UV4 cells, due to defective DNA cross-link removal [10], it nevertheless suggests that adequately high oxic/hypoxic sensitivity ratios can be achieved in vivo.

For efficient initial reduction to the nitro radical anion, the reduction potential of the prodrug needs to be within the range of the widespread mammalian reductases noted above (preferably around −350 mV). Despite the effectiveness of **2** in the above UV4 CHO model, with an E(1) of −520 mV it was less effective in a more robust cell model (oxic/hypoxic ratio of about 3-fold) [9]. Increasing the reduction potential of the nitrophenyl mustard **3** by the addition of more electron-withdrawing substituents gave compound **4**, with an E(1) of −341 mV and a oxic/hypoxic cell inhibition ratio of 58 [11], and this compound was the most effective of a series of regioisomers explored in an SAR (structure-activity relationship) study [12] (Figure 2).

Further development of this approach [13] using more efficient and more soluble leaving groups on the mustard, resulted in compound **5**, which showed improved hypoxic potency (half-maximal inhibitory concentration: IC_50_ 0.28 µM) and hypoxic/oxic cell ratio (133-fold). The corresponding phosphate (**6**), designated PR-104, was taken to Phase II clinal trials by Proacta Inc in 2006, and its major mechanism of action after activation was confirmed to be DNA cross-linking [14]. Hong et al. [15] undertook a detailed study concerning the “bystander effects” of PR-104, using 3-dimensional cell cultures. They concluded that the results could only be explained by the presence of much less polar metabolites than the initial ones, due to exchange of the Br and OSO_2_Me units by chloride to give the more rapidly diffusible metabolites **7** and **8**. In a mechanistic study [16], the levels of these active metabolites correlated well with the change in DNA reduced crosslinks or monoadducts formed. Increased cytotoxicity of PR-104 was seen in HCT116 cells expressing methionine synthase reductase, novel diflavin oxidoreductase 1 and inducible nitric-oxide synthase, identifying these as effective metabolisers of the prodrug [17]. In a phase I/II clinical trial of PR-104 in patients with relapsed/refractory acute myeloid leukaemia, 32% of patients who received 3 g/m^2^ or 4 g/m^2^ had a response, with a measurable decrease of the proportions of hypoxic cells present [18]. A spatially resolved pharmacokinetic/pharmacodynamic (SR-PK/PD) model for PR-104, using both single cell suspensions to determine relationships between PR-104A metabolism and clonogenic cell killing, and multicellular layer (MCL) cultures to measure tissue diffusion coefficients, suggested that “bystander effects” contributed 30 and 50% of PR-104 activity in SiHa and HCT116 tumours, respectively [19].

Unpredictable variable toxicity in the Phase II trials stopped further development of PR-104. This toxicity was later shown [20] to be due to aerobic activation by the aldo-keto reductase (AKR1C3) enzyme, not previously recognised as a nitroreductase. Later development of the class resulted in the related compound CP-506 (**9**), which did not have this liability and was shown to effectively decrease the hypoxic fraction and inhibit the growth of a wide range of hypoxic xenografted tumours [21]. This compound is currently in Phase II clinical trial [22].

## 3. Prodrugs Activated by Fragmentation

A second approach to hypoxia-activated prodrugs is the use of a variety of different nitroheterocyclic or nitrophenyl “triggers”, where the electron release on reduction of the nitro group results in fragmentation the linker to release a cytotoxic effector (Figure 3). This approach encompasses a wide variety of cytotoxins of differing mechanisms.

This more “modular” design is very versatile, utilising nitrobenzyl, nitrophenyl and a wide variety of nitroheterocyclic trigger units. These have varying reduction potentials and lipophilicity and use effectors encompassing a wide range of different cytotoxic mechanisms when released.

### 3.1. 4-Nitrobenzyl Triggers

The most widely used nitrobenzyl trigger in prodrugs has been the 4-nitrosubstituted isomer, due partly to its overall higher one-electron reduction potentials (e.g., −356 mV for 4-nitroacetophenone (**10**) [23]. Ge et al. [24] reported the 4-nitrobenzyl-triggered hypoxia-activated nitrosourea prodrug NBGNU (**11**), which was designed to induce DNA interstrand crosslinks following reduction, by inhibiting O6-alkylguanine-DNA alkyltransferase, an enzyme which has the ability to suppress the chemotherapeutic effect of chloroethylnitrosoureas by removing drug-induced alkyl groups from the O6-site of guanine. In human glioma SF763 cells in culture, **11** was about 5-fold more potent under hypoxic compared to oxic cultures (IC_50_s 580 and 126 µM, respectively), with much higher levels of dG-dCd crosslinks in human brain glioma SF126 than cells treated with nimustine (**12**) under hypoxia (Figure 4).

Al-Hilal et al. [25] reported their design and evaluation of a 4-nitrobenzyl-triggered prodrug (**13**) of fasudil (**14**), a known inhibitor of the Rho regulator ROCK. Prodrug **13** showed an attenuated IC_50_ of 6.8 µM compared with that of fasudil itself (IC_50_ 0.05 µM) against cells cultured from patients with pulmonary arterial hypertension and grown in air. When the cells were grown under hypoxia, **13** showed an IC_50_ of 0.05 µM, undergoing fragmentation to give a mixture of primarily **15** plus some fasudil, with an overall IC_50_ of 0.03 µM.

Liu et al. [26] designed a novel prodrug (FDU-DB-NO2: **16**) that on bioreductive fragmentation released not only the antimetabolite floxuridine (**17**), but simultaneously generated the fluorescent dye 4′-(diethylamino)-1,1′-biphenyl-2-carboxylate (**18**); the latter for real-time tracking of drug release. The prodrug **16** was evaluated under normoxic and hypoxic cultures of MCF-7 breast cancer and MCG-803 human gastric cancer cells, where it showed good activity (Figure 4). In severe hypoxia, the prodrug was highly effective in suppressing ROCK activity in pulmonary arterial smooth muscle and endothelial cells but was less effective in MCF-7 xenografts in nude mice.

Yeoh et al. [27] reported the preparation and evaluation of the bis(4-nitrophenyl) prodrug (**19**) of the anti-bacterial antibacterial peptoid mimic **20** [28] against cultures of *S. aureus* and *E. coli*. The prodrug was inactive with a minimum inhibitory concentration (MIC) >256 µg/mL in both strains, but when reduced (chemically, by SnCl_2_) showed MICs of 2–4 µg/mL. Luo et al. [29] developed a dual-release drug (**21**), which simultaneously generated YCH-1 hemisuccinate, which rapidly cyclised to give the HIF-1α expression inhibitor YCH-1 (**22**) and doxorubicin (**23**). In human adenocarcinoma A549 cells the prodrug **21** had an IC_50_ of 9.6 μM compared with free doxorubicin (IC_50_ 4.3 μM) but was considerably more cytotoxic (IC_50_ 1.2 μM) under hypoxia. Treatment of A549-tumor-bearing mice with doxorubicin (**23**) (2.9 mg/kg × 4 daily) or 21 (1.5 mg/kg × 4 daily) showed a 3-fold slower tumour growth with prodrug **21** than with doxorubicin (Figure 5).

4-Nitrobenzyl triggers have also been used in cases where the chemistry of the rest of the generates fragmentation following nitro-reduction. The prodrug **24** fragments preferentially under hypoxia to release the O6-alkylguanine-DNA alkyltransferase (AGT) inhibitor **25** [30]. In EMT6 cells under hypoxia **24** was efficiently converted to **25** and was able to significantly enhance the cytotoxic effect of alkylators. Zhu et al. [31] also prepared the isomeric prodrug **26** which was similarly activated under hypoxia in EMT6 cells >60-fold faster under hypoxic compared to normoxic culture conditions. Noting the effectiveness of chloroethylureas on leukaemia and lymphomas, Sartorelli’s group developed a number of 4-nitrobenzyl prodrugs of this DNA-ethylating agent. The 4-nitrobenzyl prodrugs **27** [32] and the more soluble **28** [33] both rapidly fragment on nitro group reduction to release the active alkylating agent **29** (chloretazine), which in turn releases the ethylating agent **30**. While both clorazetine and the clinical alkylating agent Carmustine (BCNU, **31**) both release 2-chloroethyl isocyanate, clorazetine is much more effective than BCNU against human erythrocytes in culture, suggesting that the co-released methyl isocyanate from clorazetine is important [34,35] (Figure 6).

### 3.2. Nitrophenyl Triggers

These have not been widely used in prodrugs due to their low reduction potential (see the development of PR-104 (**6**) above). 4-Nitrobenzoic acid has an E(1) of −425 mV [6], and Francisco da Silva et al. [36] measured reduction potentials of around −1000 mV for a series of 4-nitrophenyl compounds. However, special cases exist. SLC-0111 (**32**) is a potent inhibitor of the carbonic anhydrases CAIX and CAXII (particularly of CAXII; Ki 4.4 nM). Nocentinii et al. [37] report the synthesis and evaluation of a number of potential hypoxia-activated prodrugs of **32** (e.g., **33**), using as trigger the 3-nitrophenyl unit in the drug, whose one-electron reduction potential is significantly raised by an attached SO_2_NH_2_ group. Compound **33** was a reasonably potent inhibitor of the enzyme (Ki of 0.21 µM against CAIX and CAXII) and also showed modest selective potency for hypoxic over oxygenated cultures of MDA-MD-231 adenocarcinoma cell line.

Sansom et al. [38] describe the design and evaluation of an intriguing prodrug (**34**) where a 2-nitrobenzyl trigger, when reduced, fragments to simultaneously release the kinase inhibitor semaxanib (**35**) together with the nitrogen mustard (**36**). The prodrug was evaluated in both aerobic and hypoxic cultures of MDA-MB-468 breast cancer cells, but no hypoxic selectivity was seen. This was likely due to low reduction potentials (measured as −509 mV for prodrug **34**) (Figure 7).

### 3.3. 2-Nitroimidazole Triggers

2-Nitroimidazole is the more widely used of the imidazole-type triggers in the fragmentation concept, due to its good hydrophilicity and its relatively high one-electron reduction potential (–243 mV for 1-methyl-2-nitro-1H-imidazole-5-carbaldehyde (**37**) and –398 mV for 1-(2-hydroxyethyl)-2-nitroimidazole (**38**)), well within the range of various reductase enzymes. The most well-studied compound of this general class is evofosfamide (TH-302) (**39**: Figure 8), in which reduction of the nitroimidazole trigger generates a reactive DNA cross-linking alkylating agent (Figure 3). O’Connor et al. [39] reported an improved synthesis of 2-nitroimidazoles and applied this to an efficient synthesis of evofosfamide. Takakusagi et al. [40] demonstrated that while monotherapy with Evofosfamide in HT29 human colorectal adenocarcinoma and SCCVI murine squamous cell carcinoma xenografts showed only modest effects, combination with ionizing radiation showed significant benefits in both models. Hong et al. [41] used a pharmacokinetic model to show that the two metabolites of Evofosfamide; bromo-isophosphoramide mustard (Br-IPM) and dichloro isophosphoramide mustard (IPM) did not diffuse far from where they were generated but were potent and effective cytotoxins. Kishimoto et al. [42] used quantitative oxygen pressure imaging to measure the change in tumour hypoxia in response to evofosfamide treatment in pancreatic ductal adenocarcinoma xenograft models, showing that evofosfamide treatment killed cells in the hypoxic region of the tumour but also improved oxygenation in the residual tumour regions, providing a rationale for combination radiation therapy.

Recent Phase I/II combination clinical trials of **39** in various cancers have been reported. Brenner et al. [43] reported a Phase II trial of bevacizumab and evofosfamide in bevacizumab-refractory glioblastoma, on the basis of induced hypoxia being an aspect of resistance to bevacizumab, but the response rate was very low (9%). Grande et al. [44] carried out a recent Phase II clinical trial in pancreatic neuroendocrine tumours of Evofosfamide and the tyrosine kinase inhibitor sunitinib, suggesting that increased sunitinib-induced tumour hypoxia might promote activation of the prodrug, but the combination had high toxicity. Tran et al. [45] used the kinase inhibitor sorafenib, but again with limited effectiveness.

Ghedira et al. [46] prepared and evaluated 2-nitroimidazole prodrugs of phosphoramide mustards targeted at chondrosarcomas, a malignant and hypoxic cartilaginous tumour. The most effective was **40** (IFC05016), which showed an oxic/hypoxic cytotoxicity ratio of 24-fold. In a later related study, Gerard et al. [47] examined analogues (e.g., **41**) where the quaternary salt group was attached to the 2-nitroimidazole trigger, but these compounds had considerably less hypoxic selectivity. It is possible that the directly linked quaternary salt significantly altered the nitro group redox potential. Penketh et al. [48] developed and evaluated the prodrug KS-119 (**42**), designed to release a very short-lived bis(sulfonyl)hydrazine DNA alkylating agent following bioreduction, focusing cytotoxic stress on radio-resistant hypoxic cancer cell populations (Figure 8).

Karnthaler-Benbakka et al. [49] prepared and evaluated 2-nitroimidazole prodrugs of the multi-kinase inhibitors sunitinib and erlotinib. The sunitinib prodrug **43** was shown to be excluded from binding to the VEGF receptor, but had little hypoxic selectivity in cell line assays, due in part to low stability. Calder et al. [50] showed that a 2-nitroimidazole prodrug (**44**) of the clinically approved lysine deacetylase (KDAC) inhibitor vorinostat released the parent drug by enzyme-mediated bioreduction at oxygen levels below 1%, suggesting a possible utility in hypoxic tumours. A later study [51] on a 2-nitroimidazole prodrug (**45**) of the more potent lysine deacetylase inhibitor panobinostat showed hypoxia-selective enzymatic release of the parent drug in cell culture and in tumour cell xenografts in hypoxic mouse xenografts (Figure 9). Ikeda et al. [52] prepared a 2-nitroimidazole prodrug (**46**) of doxorubicin, which on enzymatic reduction generated the cyclic fragment and free doxorubicin (**23**). In colon 26 xenografts in mice, the prodrug was as effective as doxorubicin and much less toxic (Figure 9).

Jin et al. [53] reported the synthesis and evaluation of two 2-nitroimidazole-based prodrugs (**47** and **48**), bearing different trigger/drug linkers to the camptothecin metabolite and topoisomerase inhibitor SN-38, noting them as promising hypoxia-selective antitumor agents (Figure 10). Huang et al. [54] designed a more complex 2-nitroimidazole prodrug (**49**) of SN38 using an alkynyl-based click chemistry approach to also attach a pegylate-based solubilising group. An initial study in HepG2 and A549 showed that **49** rapidly released the active drug efficiently under hypoxic conditions. Dragovich et al. [55] described 2-nitrobenzimidazole prodrugs (e.g., **50**) attached to the reactive amine of the pyrrolobenzodiazepine active metabolite of the topoisomerase I inhibitor S38 (Figure 10). The prodrug was an efficient substrate for cytochrome P450-reductase, releasing S38 under hypoxic conditions in human NCI460 non-small-cell lung cancer cells in culture (oxic IC_50_ 232 nM, hypoxic IC_50_ 17 nM), but related dimeric compounds were not substrates for the enzyme.

Reports on the development of hypoxia-activated prodrugs of kinase inhibitors have also been published. Wong et al. [56] reported a novel 2-nitroimidazole prodrug (**51**) of the known [57] DNA-dependent protein kinase (DNA-PK) inhibitor IC8736 (**52**), which enables the repair of radiation-induced DNA double strand breaks. The prodrug was markedly less cytotoxic than **52**, with molecular modelling studies suggesting it is unable to occupy one of the predicted DNA-PK binding modes accessible to **52**. Bielec et al. The author in [58] reported 2-nitroimidazole-based prodrugs of the ALK/c-MET kinase inhibitor crizotinib, employing both carbamoyl (**53**) and alkyl (**54**) linkers to block the 2-aminopyridine function that is essential for the binding of crizotinib to the kinase. The carbamoyl-linked prodrug **53** was the more selective of the two, with an IC_50_s for ALK kinase >3000-fold and c-Met kinase 6-fold higher than that for crizotinib, respectively (Figure 10).

Dickson et al. [59] prepared and evaluated 2-nitroimidazole prodrugs of the 4-benzylphthalazin-1(2H)-one portion of the clinical poly(ADP-ribose)polymerase (PARP) inhibitor olaparib to investigate their value as hypoxia-activated inhibitors of PARP. Prodrug **55** was 160-fold less potent than the parent **56** on cells but did not release **56** on irradiation. In contrast, the phenol-linked prodrug **57** was only 7-fold less potent than the known PARP inhibitor parent **58** (NU-1025) but was effectively converted to it by irradiation. The work identified a useful prodrug strategy for future exploration.

The hypoxia-inducible carbonic anhydrase CAXII contributes to supporting the acidic extracellular microenvironment found in hypoxic cancers. Anduran et al. [60] prepared a number of prodrugs of the CAXII inhibitor benzenesulfonamide. The best of these prodrugs (**59**) showed modest (2.5-fold) selectivity towards hypoxic over oxic HCT29 and HCT116 colorectal cancer cell lines in culture. Hay et al. [61] compared 2- (**60**) and 5-nitroimidazole prodrugs (**61**) that release *N*,*N*-bis(2-chloroethyl)amine (**62**) on nitro-reduction. The prodrugs were considerably less cytotoxic than **62** in AA8 cells in air (IC_50_s 590, 1000 and 110 nM, respectively), but were only moderately hypoxia-selective in clonogenic assays (CT_10_ values of >1.8 and 1.7 mM-h, respectively) (Figure 11).

### 3.4. 4-Nitroimidazole Triggers

4-Nitroimidazoles have relatively low reduction potentials (a published example cites −577 mV [62]) and they have not been used extensively as triggers, However, the major example of the use of a 4-nitroimidazole trigger is tarloxotinib bromide (**63**), which on one-electron reduction releases the irreversible EGFR kinase inhibitor **64**. The prodrug showed potent initial activity against a range of HER-family oncogenes [63] and in various HER2-mutant cell lines, with HER-3 over-expression being the major mechanism of resistance. Tarloxotinib has undergone several Phase I/II clinical trials [64,65] against various types of cancer with the results being mixed and was recently reported [66] to be in an ongoing multicentre phase II study evaluating its efficacy in patients with oncogenic alterations in EGFR, ERBB2, ERBB4, or NRG1 genes (Figure 12).

### 3.5. Other Nitroheterocyclic Triggers

Winn [67] reported, among other analogues, nitrofuran (**65**) and nitrothiophene (**66**) prodrugs of phenstatin (**67**), as potent inhibitors of tublin polymerisation. They were much less toxic (IC_50_ > 20 µM) under oxic conditions than phenstatin, and when treated with NADPH cytochrome P450 oxidoreductase under anoxic conditions for 24 h, were fully cleaved. Thompson et al. [68] prepared and evaluated a number of 5-nitroheterocyclic prodrugs, including **68** and **69** of the closely related tubulin inhibitor combretastatin A4 (**70**) (Figure 13). They showed that the gem-dimethyl substitution increased the rate and efficiency of reductive elimination (a 10-fold decrease in radical half-life to 130 ms for **68** over the monomethyl analogue). The prodrug was stable in human adenocarcinoma A549 cells in air, but rapidly released **70** at oxygen concentrations of <0.1%. The best-performing compound in this study was the 2-nitrofuran **69**, which showed IC_50_s of 2.2 µM and 0.05 µM, respectively, against oxygenated and hypoxic cultures of human A549 adenocarcinoma cells in culture; a selectivity of 44-fold. In an in vivo bioluminescence imaging study in a 4T1 syngeneic mouse breast tumour model, **69** induced vascular disruption. Winn et al. [69] also reported the synthesis of the combretastatin prodrug **68**, and showed it was superior to related nor-methyl and mono-methyl congeners, being inactive as an inhibitor of tubulin polymerization (IC_50_ > 20 μM), and a 41.5-fold hypoxia-selective activation in the A549 cell line.

Sun et al. [70] reported the synthesis and evaluation of bis-pyridinium-based prodrugs, of which the nitrofuran analogue **71** had the most negative reduction potential (E = −1.56 V) and was the most potent, with IC_50_s of about 100 µM in HepG2 hepatocellular carcinoma and 3LL Lewis lung carcinoma cells under hypoxia (Figure 13).

In a comprehensive study, Tercel et al. [71] explored a range of nitroheterocyclic prodrugs, including **72–77**, for their hypoxia-activated release of mechlorethamine (**78**), determining their one-electron reduction potentials and their ratios of potency (measured as µM) in EMT6 cells cultured under oxic and hypoxic conditions) (Table 1). The “stand-out” compound was **73**, with a hypoxic cytotoxicity ration of 2500 in murine mammary carcinoma EMTC cells in culture (Figure 14).

### 3.6. Nitrobenzo[e]indole Prodrugs

The racemic 5-aminoduocarmycin analogue **79**, which binds in the minor groove of DNA in AT-rich regions and selectively alkylates the N3 of adenines, preventing DNA unwinding on replication, was shown to be a potent cytotoxin (IC_50_ 0.43 nM in AA8 cells after 4 h exposure in air) [72]. The corresponding nitro derivative **80** has an IC_50_ of 1600 µM in the same assay, making the class of potential interest as hypoxia-activated prodrugs. Wilson et al. [73] showed that the high air/anoxia potency differentials (300–500 fold) of the nitrochloromethylindolines is due to their rapid and selective alkylation of adenine N3 in the minor groove of DNA by the immediate amino reduction product. The S-enantiomers in this class were shown to be considerably (about 65-fold) more cytotoxic than the R-enantiomers in cell line assays [74], prompting Heinrich et al. [75] to develop a new six-step enantiomeric synthesis, from 2-naphthol (**81**), of the core (S)-(2,3-dihydro-1H-benzo[e]indol-1-yl) methanol (**82**) which could then be 5-nitrated (Figure 15). However, **80** showed little oxic/hypoxic selectivity towards cancer cell lines in culture, which may be due to its rather low one-electron reduction potential of −512 mV and its low solubility. A later study [76] of a series of analogues with electron-withdrawing 7-substituents and a solubilising side chain were more promising; the 7-SO_2_NH_2_ analogue **83**, with an E(1) of −390 mV, showed oxic/hypoxic ratios of 240-fold and 180-fold against C33A cervix and PC3 prostate cancer cell lines, respectively, in culture, even as the racemate. Building on the properties of **83**, Tercel et al. [77] reported an additional novel series of 5-nitroduocarmycin-like 1,2-dihydro-3H-benzo[e]indoles bearing strongly and weakly basic sidechains on both sides of the molecule. The best was **84**, which was very hypoxia selective in vitro and was soluble enough to allow formulation for in vivo studies, but which showed poor in vivo activity in SKOV3 and HT29 tumour explants, possibly due to slow diffusion. In a later study [78], the phosphate “pre-prodrug” **85** was explored against mice with implanted SiHa tumours. They were treated first with radiation to kill the oxygenated cells, followed by a 42 mmol/kg dose of phosphate prodrug **85** to treat the remaining (ca. 1%) of hypoxic cells. This protocol achieved a 128-day growth delay compared with 16 days for radiation alone (Figure 15).

Tercel et al. [79] later undertook an extensive structure-activity study of the whole class of 5-nitroduocarmycin hypoxia-selective prodrugs and concluded that those showing the highest hypoxic selectivity in various in vitro studies were associated with analogues bearing the more basic side chains (e.g., compound **84**), which showed good deactivation and high oxic/hypoxic ratios in cell line assays (for example, 50-fold in HT29 human colorectal adenocarcinoma cells and 27-fold in SiHa cervical cancer cells in culture). Treatment of mice bearing A2780 ovarian tumours with a combination of gemcitabine and the phosphate analogue **85** resulted in complete tumour regression, with the animals remaining tumour-free at 100 days. A later paper [80] reported an improved 10-step synthesis of a closely related phosphate pre-prodrug of this series (**86**), needing only two chromatographic-based purification steps, with an overall yield of over 40%. A detailed pharmacokinetic/pharmacodynamic modelling study on the related analogue **87** was later carried out [81], using measured diffusion rates of the intact drug through multicellular layer cell cultures and clonogenic cell killing in multicellular spheroid co-cultures of cells transfected with cytochrome P450 oxidoreductase (POR) or E. coli nitroreductase NfsA. This showed that **87** generated a highly efficient bystander effect through local diffusion of an active metabolite in tumour tissue (Figure 16).

The duocarmycin nucleus has also featured in other types of hypoxia-activated prodrugs. Synthesis of the 2-nitroimidazole carbamate (**88**) [82] and its evaluation against SKOV human ovarian cancer cells in a 18-hr exposure showed an IC_50_ of 75 nM, compared to a value of 1.1 nM for **79**. Similar preparation and evaluation of a range of substituted 4-nitrobenzyl carbamate prodrugs [83] against wild-type or NTR-transduced cell lines showed superior cytotoxicity toward the latter (e.g., for compound **89**, IC_50_ ratios of about 10-fold in towards NTR-transduced WiDR and V79 cell lines).

Exploration has also extended to changes in the leaving group. Ashoorzadeh et al. The author in [84] explored the effects of sulfonate, rather than chloride leaving groups in the nitroduocarmycin series. The results were variable with the nature of the leaving group; many showed little effect, but the best of a series of compounds were **90–92**, with IC_50_ oxic/hypoxic ratios of 39-, 146- and 246-fold after 4-hr exposures in HT29 human adenocarcinoma, and SiHa human cervical cancer and SKOV3 cell lines in culture, respectively. Stevenson et al. [85] studied analogues containing OSO_2_Ph leaving groups with a range of different 5-substituents. The best of these were **93–95**, with oxic/hypoxic selectivity ratios (IC_50_s) of 94-, 1370- and 1010-fold, respectively in HT29 human colorectal adenocarcinoma cells in culture. This high selectivity was attributed to the bulkier and easily eliminated benzylsulfonate leaving groups. Stevenson et al. [86] also evaluated the effect of bromide leaving groups on the bioactivity of a series of 5-nitroduocarmycins. There was an overall trend for compounds with electron-withdrawing groups at the 5-position to show higher oxic/hypoxic cytotoxicity differentials, suggesting the concomitant raising of the 5-nitro group reduction potentials favour more rapid activation in hypoxic cell line assays. Oxic/hypoxic ratios in HT29 cells after 4-hr exposure of 94-, 1370- and 1014-fold for compounds **96**, **97** and **98, respectively**. Overall, the duocarmycin motif has proved to be a very successful one for the generation of highly selective hypoxia-activated prodrugs, both in vitro and in vivo (Figure 16).

## 4. Conclusions

Many tumours generate hypoxic cores due to outgrowing their (often chaotic) internal blood vessel network, and these cells show increased resistance to radiation and chemotherapy by being both hypoxic and out of cycle. This has provided a powerful impetus for the development of hypoxia-activated prodrugs capable of specifically targeting such hypoxic cells, by exploiting the very lack of oxygen to allow activation of a prodrug specifically in such hypoxic cells. The vast majority of these prodrugs, discussed here, rely on nitroaromatic “triggers” with one-electron reduction potentials between ca. −250 to −450 mV, to restrict prodrug activation and generation of a potent cytotoxin to severely hypoxic cells. A great deal of work has been carried out, and much chemical ingenuity has been shown, in developing a very varied array of nitroaromatic triggers married to an equally wide range of masked toxins. Many of these prodrugs show very high hypoxic selectivity in front-line cell culture assays, and in many cases excellent tumour control in mouse models. Despite this, there has been limited clinical success to date, with no compounds approved and only two: CP-506 (**9**) and tarloxotinib (**63**), currently in clinical trial. The major reason suggested for this (Su et al. [87], Anduran et al. [88]) is a lack of widely accessible and reliable biomarkers for the real-time presence of tumour hypoxia in the clinic. Spiegelberg et al. [89] also noted the overall lack of therapeutic success when bringing various hypoxia-activated prodrugs into the clinic, despite the accepted importance of tumour hypoxia in many cancers and promising early clinical data. They propose a biomarker-stratified enriched Phase III study design, in which only biomarker (hypoxia)-positive patients are randomized between standard treatment and the combination of standard treatment with a hypoxia-activated prodrug, with a Phase II study to evaluate biomarkers.

## Figures and Tables

**Figure 1 pharmaceuticals-15-00187-f001:**
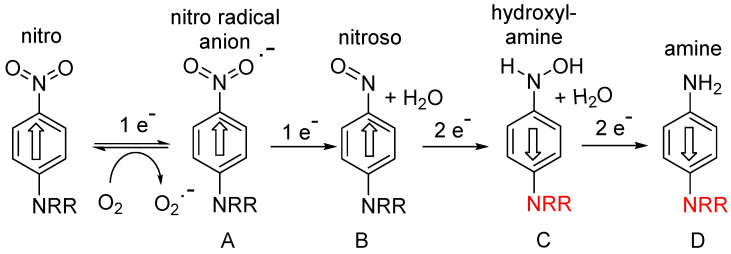
Stepwise reductive activation of nitroaromatic hypoxia-activated prodrugs (after refs. [5,6]).

**Figure 2 pharmaceuticals-15-00187-f002:**
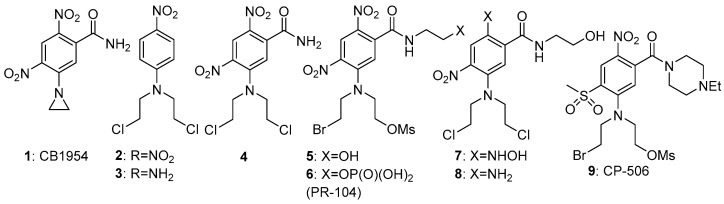
Evolution of PR-104 (**6**) from CB-1954 (**1**).

**Figure 3 pharmaceuticals-15-00187-f003:**
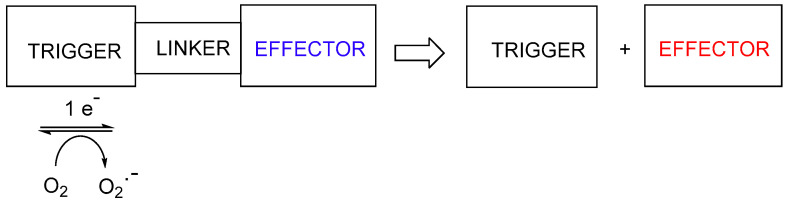
Reductive fragmentation of hypoxia-activated cytotoxins.

**Figure 4 pharmaceuticals-15-00187-f004:**
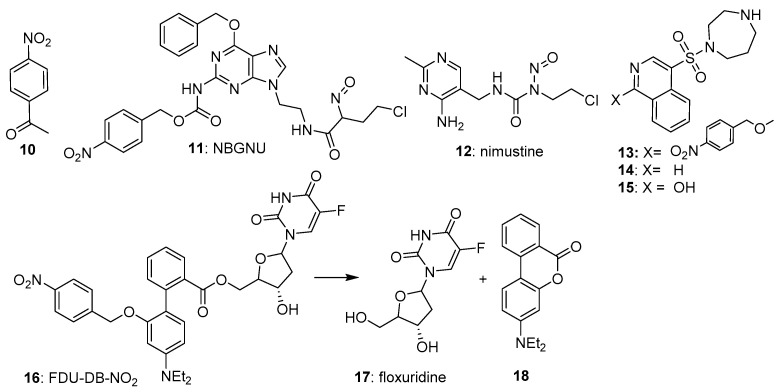
4-Nitrobenzyl-based prodrugs **10**–**16**.

**Figure 5 pharmaceuticals-15-00187-f005:**
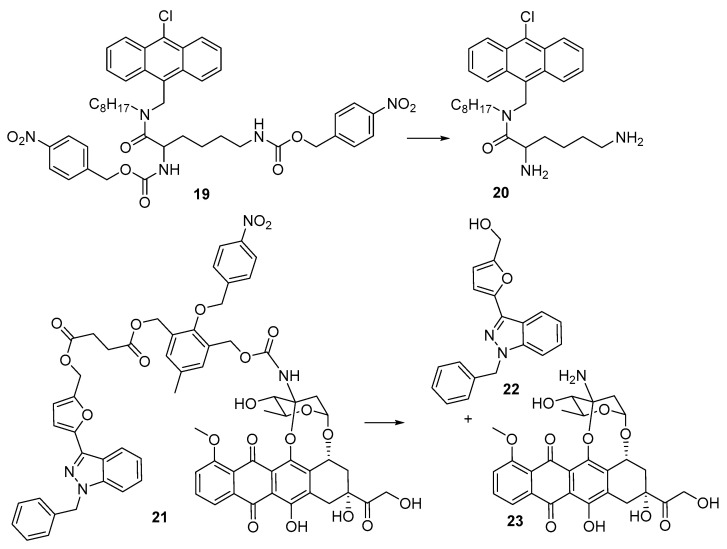
4-Nitrobenzyl-based prodrugs **19** and **21**.

**Figure 6 pharmaceuticals-15-00187-f006:**
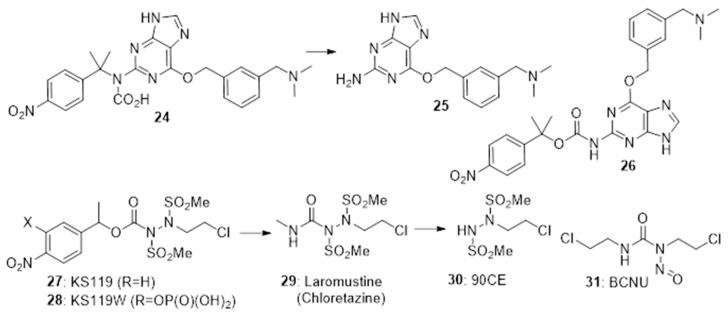
4-Nitrobenzyl-based prodrugs **24**, **26**, **27** and **28**.

**Figure 7 pharmaceuticals-15-00187-f007:**
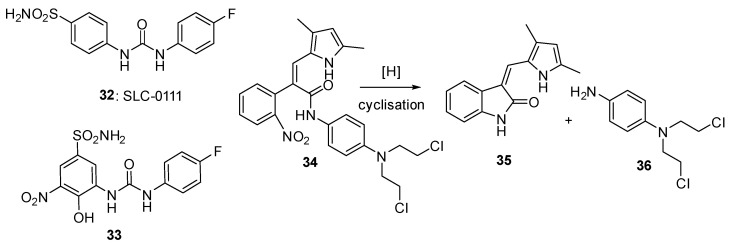
Nitrophenyl-based prodrugs **33** and **34**.

**Figure 8 pharmaceuticals-15-00187-f008:**
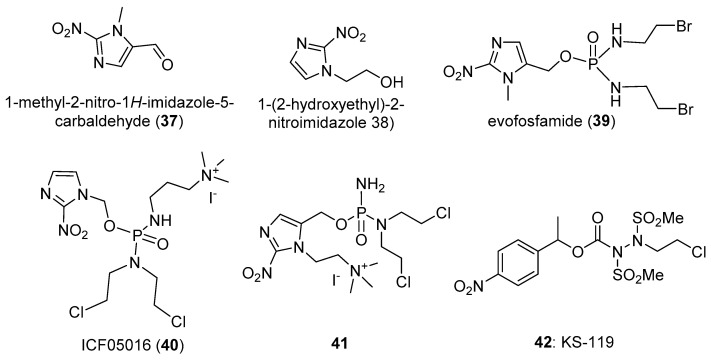
Prodrugs with 2-nitrobenzimidazole triggers releasing DNA-alkylating agents.

**Figure 9 pharmaceuticals-15-00187-f009:**
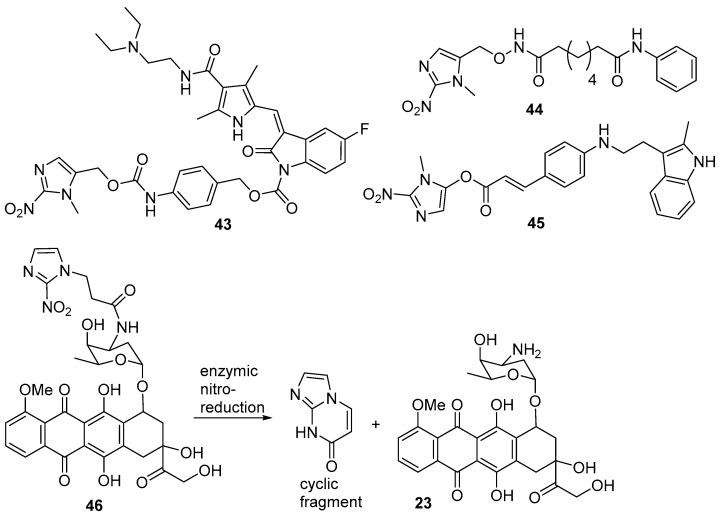
2-Nitroimidazole prodrugs **43**–**46**.

**Figure 10 pharmaceuticals-15-00187-f010:**
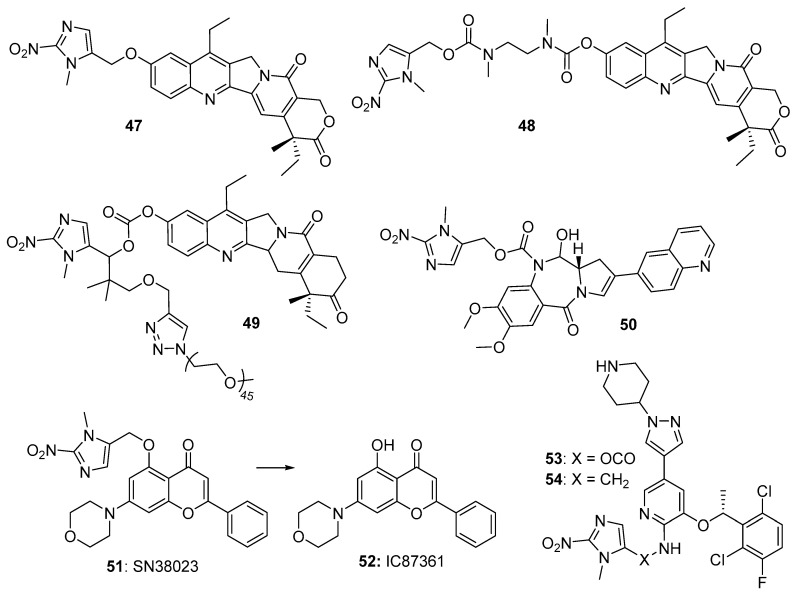
2-Nitroimidazole prodrugs **47**–**51**, **53**, **54**.

**Figure 11 pharmaceuticals-15-00187-f011:**
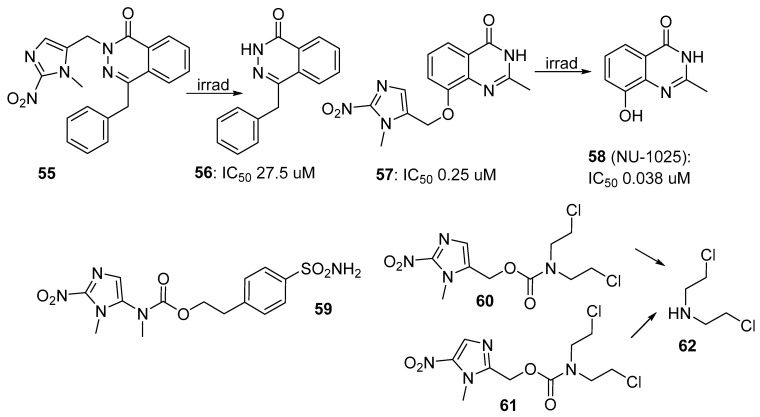
2-Nitroimidazole prodrugs **55**, **57**, **59** and **61**.

**Figure 12 pharmaceuticals-15-00187-f012:**
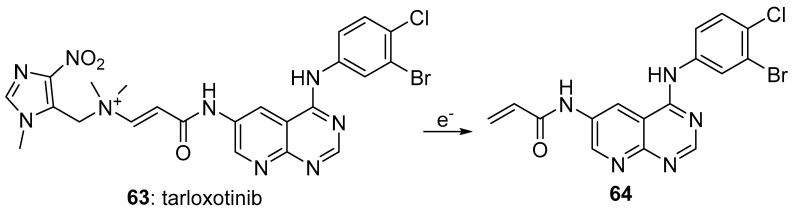
The 4-nitroimidazole clinical candidate prodrugs tarloxotinib.

**Figure 13 pharmaceuticals-15-00187-f013:**
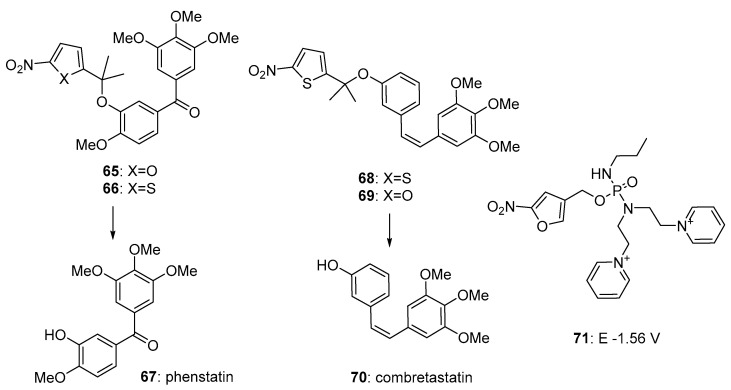
Prodrugs with miscellaneous nitroheterocyclic triggers.

**Figure 14 pharmaceuticals-15-00187-f014:**
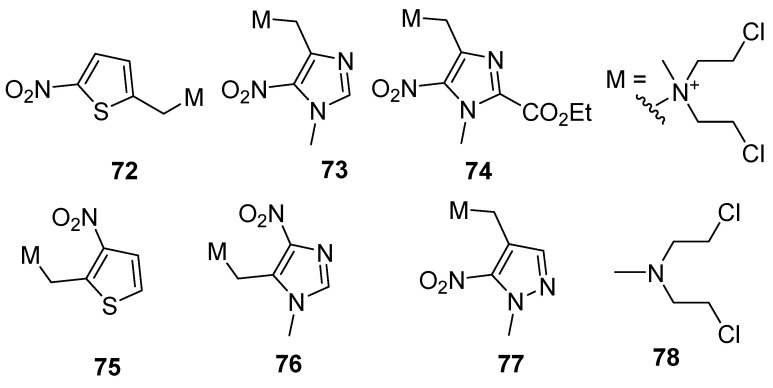
Various nitroheterocyclic prodrugs of mechlorethamine.

**Figure 15 pharmaceuticals-15-00187-f015:**
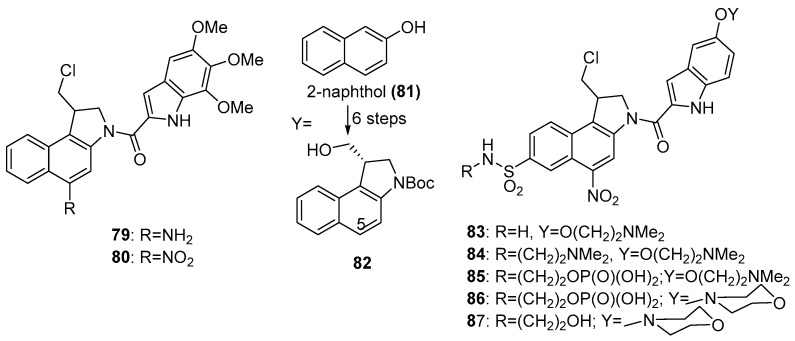
Duocarmycin analogues.

**Figure 16 pharmaceuticals-15-00187-f016:**
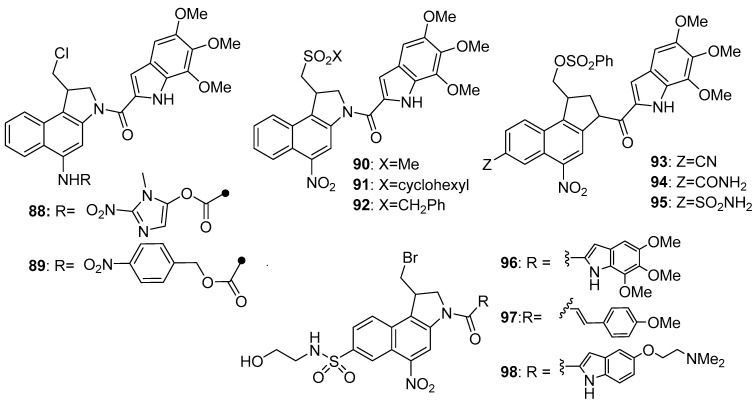
Duocarmycin analogues with different leaving groups.

**Table 1 pharmaceuticals-15-00187-t001:** Data for nitroarylmethyl quaternary nitrogen mustards.

Cmpd	E1 (mV)	C_10_ Air (µM)	C_10_ N_2_ (µM)	HR
**72**	−277	22	1.3	16
**73**	−287	3450	2.2	2500
**74**	−344	30	4.5	14
**75**	−361	30	6	6
**76**	−397	57	1.2	46
**77**	−500	245	18	16
**78**		1.0	0.85	1.17

One-electron reduction potentials, E(1), determined by pulse radiolysis. C_10_: Drug concentration to reduce surviving fraction to 10% of controls in EMT6 breast cancer cells. Values are the means for 2–3 experiments. HR = Hypoxic Cytotoxicity Ratio (C_10_ air/C_10_ N_2_). HN2 = mechlorethamine.

## Data Availability

Data sharing not applicable.

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
