# Peer review of "Nitroaromatic Hypoxia-Activated Prodrugs for Cancer Therapy"

_pharmaceuticals, 2022, doi:10.3390/ph15020187_

Round 1

Reviewer 1 Report

see an attachment

Author Response

Reviewer 1 did not request any changes. 

Reviewer 2 Report

In this work, the Author discusses nitroaromatic “hypoxia-activated prodrugs” designed to undergo enzyme-based nitro group reduction in hypoxic regions, to generate active drugs. Nitro-based prodrugs are classed by the nature of their nitroaromatic trigger units. The work contains many valuable examples of such compounds. I consider the strengths of the work to be: an interesting topic in the context of new anticancer therapies, a rich range of literature (most of the papers published after 2010), good readability. However, I have some comments / suggestions:

  1. I recommend numbering the individual chapters and subsections to obtain better clarity of the work.
  2. Lines 7, 21: Please check if the oxygen level unit is correct. Shouldn't it be ‘mmHg’?
  3. Line 74: Explain the abbreviation SAR. The same for the IC50 and the MIC that appear further. In turn, HAP, an abbreviation for hypoxia-activated prodrugs, that appears only once in Conclusions (line 469), could be used earlier or omitted.
  4. Line 98: You should omit ‘human’ at this point. Clinical trials are designed to be conducted in humans.
  5. The quality of Figure 1 and its caption as well as Figure 7 should be changed.
  6. Lines 387, 392, 419: Make sure about the references [77], [78], [82].
  7. ‘P450’ comes in several forms at work (e.g., line 41: P-450, line 271: P 450, line 329: P450). Also pay attention to the notation as 18-hour (line 420), 4-hr (line 430), etc.
  8. Figure 8: Above the arrow, replace [H} with [H].
  9. In general, the manuscript should be carefully rechecked for typos and errors, e.g.:
  • Correct the typo in the words ‘linited’ (line 231), ‘S-38’ (line 270), ‘0f” (line 356).
  • Lines 121, 205, 306, 406: Pay attention to the parentheses.
  • Line 144: A double dot at the end of a sentence.
  • Line 180: Delete ‘drugs’ in this sentence.
  • Line 195: It should be ‘This was likely due to…’.
  • Line 300: Double ‘in’ in this sentence.
  • Line 313: ‘However’ in lowercase.
  • Line 333: Replace msec to ms.
  • Line 414: E. coli should be in italics.
  • There are some minor errors in the notation of literature that should be corrected (e.g., lines 521, 524, 530, 534, 562, 601, 654, 905).

Author Response

Reviewer 2 correctly points out a number of fairly minor errors. These have all been corrected (below). I have also numbered the individual chapters and subsections as suggested.

I recommend numbering the individual chapters and subsections to obtain better clarity of the work. Done

Lines 7, 21: Please check if the oxygen level unit is correct. Shouldn't it be ‘mmHg’? Corrected.

Line 74: Explain the abbreviation SAR. The same for the IC50 and the MIC that appear further. In turn, HAP, an abbreviation for hypoxia-activated prodrugs, that appears only once in Conclusions (line 469), could be used earlier or omitted. HAP now omitted; IC50 and MIC now defined at first use (lines 78 and line 149 respectively).

Line 98: You should omit ‘human’ at this point. Clinical trials are designed to be conducted in humans. Corrected; omitted.

The quality of Figure 1 and its caption as well as Figure 7 should be changed.  Due to this and comments from other reviewers, and realizing that (old) figures 2 and 3 (now figs 1 and 2) provide the same information, this figure has been removed from the manuscript.

Lines 387, 392, 419: Make sure about the references [77], [78], [82]. These are correct; ref 77 discusses a cationic sulfonamide sidechain, ref 78 a phosphate side chain, ref 83 a nitrocarbamate sidechain, all off the same nitrobenzindoline core unit.

‘P450’ comes in several forms at work (e.g., line 41: P-450, line 271: P 450, line 329: P450). Also pay attention to the notation as 18-hour (line 420), 4-hr (line 430), etc. Now corrected; P450 and 4hr

Figure 8: Above the arrow, replace [H} with [H]. Done

In general, the manuscript should be carefully rechecked for typos and errors, e.g.:

Correct the typo in the words ‘linited’ (line 231), ‘S-38’ (line 270), ‘0f” (line 356). All corrected and a complete recheck done

Lines 121, 205, 306, 406: Pay attention to the parentheses. Done (and checked throughout)

Line 144: A double dot at the end of a sentence. Done

Line 180: Delete ‘drugs’ in this sentence. Done

Line 195: It should be ‘This was likely due to…’. Done

Line 300: Double ‘in’ in this sentence. Done

Line 313: ‘However’ in lowercase. Done

Line 333: Replace msec to ms. Done

Line 414: E. coli should be in italics. Done

There are some minor errors in the notation of literature that should be corrected (e.g., lines 521, 524, 530, 534, 562, 601, 654, 905). All checked

Reviewer 3 Report

See Attached

Author Response

Response to Reviewer 3 comments

Figure one is very confusing. They don't describe it with a figure legend at all. Is the active drug which is still linked to the oxygen sensor active or does it have to be removed from the sensor to be active? Are they trying to make a point with the negative signs that are darker on the right than they are on the left? This figure needs to be changed and or a very detailed figure legend needs to be added. And the resolution and the spacing and the capitalization in the figure or not good. This is confusing: (e.g., NADPH: cytochrome P-450 reductase, ferredoxin: NADP+ reductase). What are you trying to say here?

Thank you for your advice. Realizing that (old) figures 2 and 3 (now figures 1 and 2) provide the same information, this original figure 1 has been removed.

Figure 2. Stepwise reductive activation of nitroaromatic hypoxia-activated prodrugs (after refs. 5, 6). What does this mean? If you are going to use this figure and you were going to show oxygen “gaining???” and electron as well as the nitro group gaining an electron you should draw out the nitro group and show where the electrons are being placed especially when you draw out the nitroso group. When going from the nitro radical anion to the nitroso group you also lose water so you should accommodate for that in the figure. Also, the mechanism for the two electron reduction to the hydroxylamine and the two electron reduction to the amino group should be outlined.

These suggested changes have been made to the (re-numbered) Figure 1.

The first one-electron step generates the transient nitro radical anion A, which can be efficiently scavenged (re-oxygenated) in a futile cycle by sub-micromolar levels of oxygen [6,7], preventing significant further metabolism in well-oxygenated normal tissue. The arrow is going in the wrong direction in the figure and is quite confusing.

This has been corrected.

This cascade greatly increases the electron density at the substituent R – This is not correct, the electron density is increased at the nitrogen atom and not then substituent.

The reviewer is correct; thank you for pointing this out. The wording has been changed to “This cascade greatly increases the electron density at the nitrogen bearing the substituent R”

The use of the Hammett constants is just throwing out words that make a reader think that the author is correct because they don't understand what that means. If you're going to use Hammett constants and put their values in the text you need to add that to the figure and demonstrate why you are even using these numbers.

I believe that the nature and use of Hammett constants to describe the electron-donating or -withdrawing ability of a substituent on an aromatic ring is sufficiently well-known to not require detailed explanation. However, I have added a section “This cascade greatly increases the electron density at the nitrogen bearing the substituent R, as shown by the Hammett substituent constants (a measure of the electron-donating or -withdrawing ability of a substituent on an aromatic ring); σp NO2 = +0.78 (electron-withdrawing); σp NH2 = -0.66 (electron-donating)”. Adding the values to Figure 1*.

This large difference in electron shift has been used to greatly enhance the cellular toxicity of nitrogen mustards as DNA crosslinking agents. Again, this is just another play on words that makes the reader think that the author fully aware of this system with activation. This needs to be better explained and referenced.

Figure 1 and the following paragraphs have been significantly changed to provide a clearer explanation.

I also believe the author needs to go through the entire manuscript and make sure to check grammar, check punctuation, follow the format for the journal guidelines and update all of the future figures and references with the comments I have provided thus far. Once that is done this reviewer is happy to look at the manuscript again but at this point it is not suitable for publication. There are many errors in changing of font, identification of compounds from the text to the figures, and it should all be thoroughly examined before re-submission.

The manuscript has been carefully re-checked and any other errors found have been corrected.
